# Novel augmented reality solution for improving health literacy around antihypertensives in people living with type 2 diabetes mellitus: protocol of a technology evaluation study

Alireza Ahmadvand,[1] Judy Drennan,[2] Jean Burgess,[3] Michele Clark,[1] David Kavanagh,[4] Kara Burns,[1,2] Sarah Howard,[5] Fleur Kelly,[6] Chris Campbell,[7] Lisa Nissen[1]

For numbered affiliations see end of article.

**Correspondence to**
Professor Lisa Nissen;
l.nissen@qut.edu.au

## ABSTRACT

**Introduction** Low health literacy is common in people with type 2 diabetes mellitus (T2DM) (up to 40%), associated with decreased self-efficacy in managing T2DM and its important complications, mainly hypertension. This study introduces, for the first time, an easy-to-use solution based on augmented reality (AR) on smartphones, to enhance health literacy around antihypertensive medicines. It assesses the feasibility of the solution for improving health literacy, oriented specifically to angiotensin II receptor blockers; embedding the health literacy improvement into the use cycle of angiotensin II receptor blockers and providing continuous access to information as a form of patient engagement.

**Methods and analysis** This is a technology evaluation study with one technology group (AR plus usual care) and one non-technology group (usual care). Both groups receive face-to-face communications with community pharmacists regarding angiotensin II receptor blockers; the technology group receive additional AR-enhanced digital consumer medicine information throughout the use of their medications. The primary outcome is the change in health literacy and the hypothesis is that the proportions of people who show high health literacy will be larger in the technology group. Mixed effects models will be used to analyse solution effectiveness on outcomes. Multiple regression models will be used to find additional variables that might affect the relationship between health literacy and the AR solution.

**Ethics and dissemination** Queensland University of Technology (QUT) Human Research Ethics Committee has approved the study as a low-risk technology evaluation study (approval number: 1700000275). Findings will be disseminated via attending scientific conferences and publishing in peer-reviewed journals. Facilitated by QUT, two press releases have been published in public media and two presentations have been made in university classrooms.

## Strengths and limitations of this study

► This study formally evaluates the new augmented reality solution from multiple perspectives, that is, perspective of the researchers, patients, providers and developers.
► Application walkthrough methodology and follow-up of participants to evaluate health literacy and self-efficacy are combined to enrich the design and formal evaluation of the new augmented reality solution.
► This study is a multidisciplinary partnership between six organisations, including academia, not-for-profit organisations, community organisations, private industries and service provider organisations.
► This study only assesses one category of antihypertensive medicines (ie, the angiotensin II receptor blockers) in people living with type 2 diabetes mellitus.
► This study is not equipped to evaluate more distal clinical outcomes regarding diabetes and high blood pressure, such as HbA1C or overall control of blood pressure.

## INTRODUCTION

Type 2 diabetes mellitus (T2DM) is a major challenge to Australia's healthcare system; by 2030, up to 3 million Australians may potentially develop the condition.[1] Hypertension has been reported in around 70% of people with known or undiagnosed T2DM.[2] Concurrent use of multiple medicines (often more than eight medicines) and limited health literacy about medicines largely limit optimal management of T2DM and hypertension.[3]

Almost 60% of adult Australians have low health literacy which significantly limits their agency when making health-related self-care decisions.[4] In people with T2DM, low health literacy is common (up to 40%), associated with limited knowledge of their condition,

and decreased self-efficacy in managing important cardio-vascular risk factors, mainly hypertension.[5]

Smartphone-based technologies, given their widespread uptake, offer an opportunity to improve health literacy via providing easy access to information, including health information. In parallel, patients use mobile technologies in diabetes management for self-care, to share information, collaborate with peers and improve their engagement with their healthcare providers, which is trusted to have positive effects on health literacy.[6] However, currently, the packages of medicines as part of the information that people living with T2DM receive contain 'static' consumer medicine information sheets that many people may not read, understand or take notice of. Additionally, the packages and their inserts offer no opportunity for continued education and increased patient engagement.

Augmented reality (AR) is a novel technology which allows supplementary digital content to be visualised as information overlaid or displayed alongside the user's view of physical objects. AR has been effective in improving people's literacy, with potentials in health education.[7] Our aim is to develop and test an AR-based solution for improving health literacy in T2DM around one guideline-endorsed family of antihypertensive medicines, that is, angiotensin II receptor blockers. Therefore, the generic and branded packages of angiotensin II receptor blockers will be enhanced using a custom-built AR app. Then, digital 'dynamic' consumer medicine information will be displayed seamlessly on top of the packages (figure 1) to provide a superior complement to improve health literacy, fostering continued education, from dispensing throughout the medicine use and refill cycle.

The short tile of the study is VALiD, the Value of Augmented reality for improving medicine Literacy in Diabetes and hypertension. VALiD aims to introduce, for the first time, an easy-to-use information technology solution, AR, to enhance condition-oriented health literacy. This study assesses the feasibility of a concept for improving health literacy, oriented specifically to angiotensin II receptor blockers in T2DM; embedding the health literacy improvement into the medicine use cycle and providing continuous access to information as a form of patient engagement. This project, therefore, addresses the important issue of low health literacy in T2DM and hypertension by:

1. Introducing a novel custom-built AR app to improve health literacy about antihypertensive medicines in T2DM.
2. Evaluating the effects of this solution on people's self-efficacy in managing their hypertension.

## METHODS AND ANALYSIS

VALiD is a technology evaluation study with one technology group (AR plus usual care) and one non-technology group (usual care). The level of group classification is the pharmacy. Every pharmacy will be divided into technology group versus non-technology group and all participants coming to either group will be in the technology or non-technology group correspondingly.

### Project plan summary

VALiD is a 16-month study with one technology and one non-technology group. Both groups receive the usual face-to-face communications with community pharmacists regarding angiotensin II receptor blockers. The technology group receive additional AR-enhanced digital consumer medicine information throughout the use of their medicine. The aims are to:

1. Establish a participatory AR design team using developers, social researchers, diabetes educators, community pharmacists and people living with T2DM and hypertension.
2. Select and define the packages of generic and branded angiotensin II receptor blockers as AR 'triggers'.

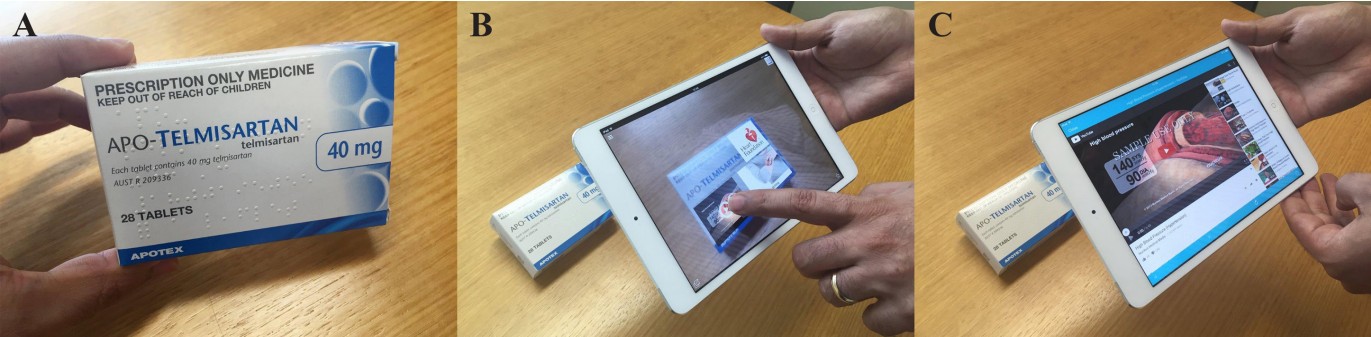

**Figure 1** An example of how the augmented reality solution in this study works: Users install specific apps and scan predefined patterns (eg, package of a medicine) via their smartphone or tablet. These patterns, technically called 'triggers', initiate the retrieval and visualisation of digital content as overlaid information. (**A**) An ordinary package of telmisartan, a frequently prescribed angiotensin II receptor blocker. (**B**) The same package, augmented by the 'Layar' app. The app retrieves extra digital information and conveniently overlays it on top of the package; a clickable YouTube video about high blood pressure, a link to the Heart Foundation's website about medication management of high blood pressure and a flip-through slideshow of the most common side effects of telmisartan are shown. (**C**) YouTube video is shown instantly after user clicks on its thumbnail. (Photo credit: Dr Kara Burns. Patient consent for photography taken by Professor Lisa Nissen)

3. Transform non-digital consumer medicine information about angiotensin II receptor blockers into creative digital formats.

4. Assign digital consumer medicine information to triggers.

5. Finalise the development of a custom-built AR app by the team (the app's trademarked name is MedAugment).

6. Evaluate usability and customise the design of the AR app and visualisation of consumer medicine information using the 'app walkthrough' user experience research method and iterative work with the AR design team.

7. Divide a group of people with T2DM and hypertension coming for their angiotensin II receptor blocker prescriptions into technology or non-technology groups (based on the pharmacy of access).

8. Train the participants in the technology group to install the AR app, scan augmented packages and use digital consumer medicine information.

9. Application walkthrough with a purposive sample of participants from the technology group.

10. Follow-up with the participants to evaluate health literacy and self-efficacy.

11. Follow-up with the participants to assess the acceptability, perceived utility and willingness for consumers and organisations to use the AR app.

12. Disseminate the results to stakeholders.

### Research environment and sites

The study field will be four selected high daily volume outpatient pharmacies of Terry White Chemmart Group pharmacies in Brisbane, Queensland. Two pharmacies serve as the technology sites while the other two are the non-technology sites. Terry White Chemmart Group equips each technology pharmacy with a research-informed pharmacist and at least one smart device with MedAugment AR app preinstalled for showing the participants how to install and use the solution during the follow-up. The training, as well as baseline and follow-up assessments happen inside the pharmacies.

### Participants and eligibility criteria

Interested people are considered eligible if they are adults living with T2DM and hypertension, aged between 30 and 60 years; own a mobile phone and data subscription; are registered under Australia's National Diabetes Services Scheme; are comfortable communicating in English; able to provide written informed consent; have used the Terry White Chemmart Group pharmacies for at least 3 months; have no intention to travel or planned procedure within 6 months of recruitment; are taking prescriptions of angiotensin II receptor blockers for at least 3 months; picking up their own prescriptions of angiotensin II receptor blockers from corresponding pharmacies; and are not enrolled in any other diabetes education programmes.

### AR app walkthrough

Using the app walkthrough method,[8] researchers will help the participants in the technology group in traversing them to reflect on the technological features and activity flows of MedAugment app; and in so doing generate qualitative insights into both the app's usability and the digital literacies of the participants. The walkthrough method involves the researchers engaging with the app, working through and closely documenting the activities and features it affords while contextualising these affordances within the app's environment.[8] Data collection during this process will take the form of screenshots or recording the walkthrough while taking field notes or audio recording reflections. If some devices do not allow for screen recording, another device will be used to capture activity on a mobile or tablet screen. The researchers undertake these walkthroughs twice:

1. The walkthrough is undertaken by the researchers independently at the commencement of the project to assess the app's usability and inform any required design refinements.

2. During the project, the researchers guide participating users through the walkthrough to capture user experiences and perspectives in a rich and naturalistic setting to gather varying perspectives as the data are collected.

### Technology group versus non-technology group comparisons

Both groups will receive face-to-face education regarding angiotensin II receptor blockers by the pharmacists as per approved professional guidelines. In addition, participants in the technology groups will also have access to the AR-enhanced education via digital interactive content in MedAugment, tailored to angiotensin II receptor blockers and retrieved by scanning the medicine packages (figure 1). We will implement the following two-part training programme at the Terry White Chemmart Group pharmacies: (1) training sessions for pharmacists on the principles of clear health communication, as well as on the use of AR as a counselling tool; (2) AR-based education to participants when it is time to refill their prescriptions, by providing digital media content about the effects of angiotensin II receptor blockers, its indication, daily dosing schedule, precautions, adverse reactions and what to do should a reaction happens.

The non-technology group will receive usual pharmacist care during the follow-up period. The care involves the pharmacist discussing every person's individual treatments, information on the chronic nature of T2DM and hypertension, and the importance of continuous therapy, with emphasis on medication adherence. At the end of the usual care period, all people will be offered the solution outlined above.

To address the inherent complexities of comparing outcome measures in a research project focused on education and health literacy, and to assess how the understanding of AR might affect health literacy or comparability of the technology versus non-technology

groups, the following major types of comparisons will be made, as appropriate:

1. Within-group comparison, in which relevant outcome measures will be assessed and compared 'within' each group 'before' and 'after' the intervention (pre-post).
2. Between-group comparison, in which relevant outcome measures will be assessed and compared between each independent group.

### Data collection and outcome measures

1. *Baseline survey at pharmacies:* Eligible participants who provide written informed consent are enrolled in the study and complete the baseline survey. Participants are given a $15.00 gift voucher after completing the baseline and each follow-up survey. Within the pharmacy, a trained interviewer will administer an in-person baseline interview using a questionnaire which includes instruments to assess these outcome measures:
   – Health literacy (primary outcome measure), assessed by Rapid Estimate of Adult Literacy in Medicine[9]; Communicative Health Literacy[10 11]; and modified version of the Test of Functional Health Literacy in Adults (s-TOFHLA).[12]
   – eHealth literacy, assessed by eHEALS (the eHealth Literacy Scale).[13]
   – Beliefs about medication, assessed by Beliefs about Medicines Questionnaire.[14]
   – Self-efficacy, assessed by short form of the Patient Activation Measure[15] and self-care ability measure in the Diabetes Care Profile.[16]
   – Self-reported medication adherence, to be assessed by a licensed version of the Morisky Medication Adherence Scale (MMAS).[17]
   – Demographic information (demographics questionnaire).
2. *Follow-up survey at 3 months and 6 months:* Participants will be scheduled for a follow-up interview at 3 and 6 months after the baseline interview (non-technology group) and the launch of the solution (technology group). The follow-up surveys will include most of the measures from the baseline survey that have the potential to change during the study period. Additional to the follow-up surveys, the acceptability, perceived utility and willingness for participants in the technology group to use the new solution are investigated.
3. *Data analytics from the AR apps at 3 and 6 months:* VALiD will use analytics from MedAugment app to get insight into participants' behaviour around the usage of the app. By capturing user data and recording traffic data (eg, number of clicks, navigation paths, the number of views and similar), VALiD will analyse user interactions and determine popularity trends for digital content.

### Sample size justification

Sample size will be n=74 in each group, assuming 80% power, a one-sided alpha of 0.05, $P_1$=40% (proportion of people with low health literacy in the non-technology group after follow-up), $P_2$=20% (proportion of people with low health literacy in the technology group after follow-up). This sample size will be increased to 82 in each group (total: 164) to account for possible dropouts, losses to follow-up and withdrawals of consent.

### Planned timetable

The main milestones of the study and their corresponding time frames are:

1. Development of MedAugment AR app: April 2017 to August 2017.
2. Creating dynamic and engaging AR regarding angiotensin II receptor blockers: April 2017 to August 2017.
3. Recruitment and training the participants and follow-up: January 2018 to April 2018.
4. Developing analytical frameworks: December 2017 to February 2018.
5. Analysing the results: May 2018 to June 2018.
6. Reporting and disseminating the results: June 2018 to July 2018.

### Analysis

The primary comparison is the changes in health literacy between the groups as a continuous variable. The hypothesis is that the proportions of people who show high health literacy will be larger in the technology groups. Mixed effects models will be used to analyse solution effectiveness on outcomes. Multiple regression models are also used to find additional variables that might affect the relationship between health literacy and the solution. As for continuous variables, means and SDs (and medians) are calculated and differences are assessed by t-tests. For categorical variables, counts and proportions of key variables of interest are reported and differences between groups are assessed by $X^2$ tests or Fisher's exact test, as appropriate.

## ETHICS AND DISSEMINATION
### Ethical clearance

The study has received approval from the Queensland University of Technology (QUT) Human Research Ethics Committee as a low-risk technology evaluation study. Before getting ethics approval, the study protocol has been peer reviewed in two rounds of selection and shortlisting by the funding organisation.

### Expected outcomes to improve cardiovascular health

VALiD is expected to extend the engagement between community pharmacists and people living with T2DM, characterised by facilitated education and therefore, improved health literacy. It improves health literacy around angiotensin II receptor blockers for the ultimate improvement of the safety of pharmacological care of hypertension in T2DM. The AR app will be developed and evaluated in a multifaceted and participatory way, involving people living with T2DM and hypertension

from the start and using a systematic approach towards the analysis of the AR app and its appropriation. The outcome of better health literacy around angiotensin II receptor blockers will be reflected as improved services for people and evidence-informed support services can be planned by partners of this project.

By leveraging the positive role of pharmacists in improving medication safety for chronic conditions[18] and mobile phones' potential to transform health service delivery,[19] this concept can be adopted for other cardio-vascular medicines in future.

## Partnerships

VALiD is based on the partnership of QUT researchers with these institutions to seek funding for larger studies and future projects: (1) NPS MedicineWise: a nationally recognised, independent, not-for-profit organisation for medicine information and decision support; (2) Diabetes Queensland: a not-for-profit community-based organisation for people living with diabetes in the state of Queensland; (3) Terry White Chemists Group: one of the largest pharmacy chains in Australia; (4) Activate Entertainment: creative developer of engaging AR content as an industry partner; and (5) Princess Alexandra Hospital Diabetes and Endocrine Clinic—one of the largest providers of services to people with diabetes. This partnership aims to expand its focus to other cardio-vascular diseases and risk factors in the future.

## Future larger studies and funding

QUT has a pioneering position in research on emerging mobile technologies for managing chronic conditions. A multidisciplinary cross-faculty team at QUT has proactively started engaging with professional organisations and developers to evaluate the effectiveness of AR on improving the care of people living with T2DM and cardiovascular conditions. The team uses concepts from health, business, information technology and creative digital media to bring multiple perspectives to problems, and encourage innovative solutions.

## Storage of data

All study data will be stored on secure Microsoft Share-Point Services websites on QUT's internal servers. The principal investigator (LN), members of the steering committee of the project (JD, JB, MC and DK) and selected research officers of the project (KB and AA) will have secure, controlled access to the full data.

## Dissemination of findings

The dissemination of findings will be via attending scientific conferences (as planned, the National Medicine Symposium 2018), and publishing the results of the study in peer-reviewed journals. Currently, two press releases have been published in public media, and two presentations have been made in two university classrooms, all facilitated by QUT.

## Intellectual property

The Deputy Vice-Chancellor (Research and Commercialisation) at QUT can collaboratively implement strategies to commercialise the research outputs from VALiD, to promote knowledge transfer and to encourage the uptake of research by end-users. This can mainly be achieved via consultancies and customised networking collaboratively with Australia's Heart Foundation, Diabetes Queensland, NPS MedicineWise, Terry White Chemmart Group, Activate Entertainment and Princess Alexandra Hospital.

MedAugment is a registered trademark in Australia.

## Acknowledgements of specific organisations or individuals

Karen Kaye, Executive Manager—Client Relations at NPS MedicineWise; Tyronne Curtis, Director and Founder of Activate Entertainment; Kaitlyn Porter, Research Scholar at School of Clinical Sciences, QUT; Stefanie Duguay, former Research Assistant at QUT's Digital Media Research Centre; and Kerry Porter, Diabetes Educator at Princess Alexandra's Hospital Diabetes and Endocrine Clinic will contribute to the implementation of the study.

**Author affiliations**
[1]School of Clinical Sciences, Faculty of Health, Queensland University of Technology, Brisbane, Queensland, Australia
[2]School of Advertising, Marketing and Public Relations, QUT Business School, Queensland University of Technology, Brisbane, Queensland, Australia
[3]Digital Media Research Centre, Queensland University of Technology, Brisbane, Queensland, Australia
[4]School of Psychology and Counselling, Faculty of Health, Queensland University of Technology, Brisbane, Queensland, Australia
[5]QUT Library, Queensland University of Technology, Brisbane, Queensland, Australia
[6]Diabetes Queensland, Brisbane, Queensland, Australia
[7]Business Management, Terry White Chemmart Group, Brisbane, Queensland, Australia

**Collaborators** Tyronne Curtis, Kaitlyn Porter, Kerry Porter, Karen Kaye, Stafanie Duguay

**Contributors** AA and LN created the first idea of the study. JD, KB, JB, MC and DK contributed to the conceptualisation, methodology, outcome measures and grant seeking for the study. FK, SH and CC contributed to the methodology of the study, regarding recruitment, AR content and field requirements. AA and LN wrote the first draft of the manuscript. All authors contributed equally to the editing and revising of the manuscript in three rounds. LN is the principal investigator, guarantor of the study and corresponding author of the manuscript.

**Funding** This study is supported by Australia's Heart Foundation under Vanguard Grant scheme 2016–2017 (funding agreement: LM 4156). QUT will contribute to the funding of this project by the provision of time from its academics and staff, for research support and the provision of physical space, expertise, computers and library resources. Terry White Chemmart Group, NPS MedicineWise, Diabetes Queensland, Activate Entertainment and Princess Alexandra Hospital provide in-kind contribution in terms of salaries of their collaborators in this project.

**Competing interests** None declared.

**Patient consent** Not required.

**Ethics approval** Queensland University of Technology Human Research Ethics Committee (approval number: 1700000275).

**Provenance and peer review** Not commissioned; externally peer reviewed.

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
