## [Reviewer comments · BMJ Open]

ARTICLE DETAILS

TITLE (PROVISIONAL)	A Novel Augmented Reality Solution for Improving Health Literacy around Anti-Hypertensives in People Living with Type 2 Diabetes Mellitus: Protocol of a Technology Evaluation Study
AUTHORS	Ahmadvand, Alireza; Drennan, Judy; Burgess, Jean; Clark, Michele; Kavanagh, David; Burns, Kara; Howard, Sarah; Kelly, Fleur; Campbell, Chris; Nissen, Lisa

VERSION 1 – REVIEW

REVIEWER	Egui Zhu Karolinska Institutet, Sweden
REVIEW RETURNED	06-Nov-2017

GENERAL COMMENTS	It is a interesting project. However, It could be a big project which you need to develop AR, apply it as well as evaluate it. It is impossible to finish in one year. My suggestions: First, It is better to make your project plan in reasonable time plan. Second, You plan to compare in two group - technology group versus non-technology. It is not good to compare with this two group if it is an education project. Especially you used the AR plus usual care. The other group is usual care. It is better to consider how to understand AR could be improve the HL or might find other group which is comparable.
--

REVIEWER	Elena Carbone University of Massachusetts Amherst USA
REVIEW RETURNED	01-Dec-2017

GENERAL COMMENTS	The study appears to focus on functional and communicative (interactive) HL; however, the CHL measure is not specified. All measures should be referenced. By my count, there are 10 surveys at baseline, which is a heavy respondent burden. That said, an e-health literacy measure (eg., eHEALS) might be of interest to assess participants' comfort and skill in using information technology for health. The planned timeline indicates that data analysis would begin in January 2018. Given the fact that it's now December, this suggests
---

	that the data are being collected now. Therefore, I realize that changes to the methods are not possible.
--	---

VERSION 1 – AUTHOR RESPONSE

□Dear Editor,

We had the pleasure of reviewing comments made by you and the reviewers on our manuscript, entitled “A Novel Augmented Reality Solution for Improving Health Literacy around Anti-Hypertensives in People Living with Type 2 Diabetes Mellitus: Protocol of a Technology Evaluation Study”. On behalf of the team of authors, we would like to thank you for giving us the chance to revise the manuscript and benefit from the opportunity of taking on-board the feedback of two experts in the field.

Our team members have tried their best to reflect on the comments, as we have found them relevant and helpful. Please find below, the detailed explanation of our responses to each and every comment by the reviewers and the changes we have made. Inside the manuscript, the changes are highlighted using dark red colour, to provide convenience to reviewers to locate the changes.

Reviewer: 1

It is a interesting project. However, It could be a big project which you need to develop AR, apply it as well as evaluate it. It is impossible to finish in one year.

Comment #1:

My suggestions: First, It is better to make your project plan in reasonable time plan.

Authors' Response #1:

Thank you for your comment and sharing the importance of our project duration. The one year duration for project was initially set to address the requirements of the funding that we received from the Heart Foundation. However, during the 3-month time that this manuscript was being peer-reviewed, our team applied for an extension to the duration of the project. Thankfully, the project duration is now 16 months, not a year, and we have changed the timetable in the manuscript accordingly to finish the project in a reasonable time frame. The 4-month extension and the revised milestones have been reflected into the revised manuscript, in the ‘Planned Timetable’ subsection of the ‘METHODS AND ANALYSIS’ section.

As a brief, the development of the app (called MedAugment™) has finished and the two rounds of application walkthrough, one with researchers and one with people living with T2DM, have also finished. The remaining phase, recruitment and follow-up, will start in early January 2018 and is planned to finish in April 2018.

Comment #2:

Second, You plan to compare in two group - technology group versus non-technology. It is not good to compare with this two group if it is an education project. Especially you used the AR plus usual care. The other group is usual care. It is better to consider how to understand AR could be improve the HL or might find other group which is comparable.

Authors' Response #2:

Thank you for your comment. We discussed your important comment in detail and we have made these changes to the manuscript, as well as the conduct of the last stage of our study:

To address the inherent complexities of comparing outcome measures in a research project focused on education and health literacy, and to assess how the understanding of AR might affect health literacy or comparability of the technology versus non-technology groups, the following major types of comparisons will be made, as appropriate:

1) within-group comparison, in which relevant outcome measures will be assessed and compared ‘within’ each group ‘before’ and ‘after’ the intervention (pre-post); and

2) between-group comparison, in which relevant outcome measures will be assessed and compared between each independent group.

Reviewer: 2

The study appears to focus on functional and communicative (interactive) HL.

Comment #3:

However, the CHL measure is not specified.

Authors' Response #3:

Thank you for your comment. We have referenced the CHL measure now, from a well-known study by Ishikawa et al. The measure is not a stand-alone, independent measure; it is a series of questions that collectively address the CHL and we will use the same series of questions in our study.

Comment #4:

All measures should be referenced.

Authors' Response #4:

Thank you for your comment. We have put the references to every measure.

Comment #5:

By my count, there are 10 surveys at baseline, which is a heavy respondent burden. That said, an e-health literacy measure (eg., eHEALS) might be of interest to assess participants' comfort and skill in using information technology for health.

Authors' Response #5:

Thank you for your comment. We discussed your important comment in detail and we have reduced the number of surveys, after adding the e-health literacy (eHEALS) measure. We found eHEALS relevant and useful. Collectively, there is no eight small surveys at baseline, not ten.

Comment #6:

The planned timeline indicates that data analysis would begin in January 2018. Given the fact that it's now December, this suggests that the data are being collected now. Therefore, I realize that changes to the methods are not possible.

Authors' Response #6:

Thank you for your comment. During the time that this manuscript was being peer-reviewed, our team applied for an extension and now, the project duration is 16 months. We have changed the timetable in the manuscript accordingly. The revised milestones have been reflected into the revised manuscript, in the 'Planned Timetable' subsection of the 'METHODS AND ANALYSIS' section. The remaining phase, recruitment and follow-up, will start in early January 2018 and is planned to finish in April 2018. Therefore, making changes to the methods has been possible.

We would like to thank you for peer-reviewing our manuscript and we look forward to receiving your feedback/decision.

Respectfully yours,

Dr Alireza Ahmadvand and Professor Lisa Nissen (on behalf of all authors)